



# Performance assessment of state-of-the-art and novel methods for remote compliance monitoring of sulphur emissions from shipping

Jörg Beecken[1,4], Andreas Weigelt[1], Simone Griesel[1], Johan Mellqvist[2], Vladimir Conde[2], Daniëlle van Dinther[3], Jan Duyzer[3], Jon Knudsen[4], Bettina Knudsen[4], Leonidas Ntziachristos[5]

[1]Federal Maritime and Hydrographic Agency (BSH), Hamburg, Germany
[2]Chalmers University of Technology, Gothenburg, Sweden
[3]Netherlands Organisation for Applied Scientific Research (TNO), Delft, the Netherlands
[4]Explicit ApS, Copenhagen, Denmark
[5]Aristotle University Thessaloniki, Thessaloniki, Greece

*Correspondence to*: Jörg Beecken (jbe@explicit.dk)

Keywords: ship exhaust, sulphur emission, emission monitoring, remote measurements, fuel sulphur content, MARPOL Annex VI, compliance monitoring

**Abstract.** Sulphur emissions to the air from sea-going and inland vessels were measured simultaneously by eight different, state-of-the-art and novel, monitoring systems during a six-week campaign at the Elbe River, at about 10 km distance to the port of Hamburg, Germany. Both, stationary, and airborne systems on unmanned aerial vehicles (UAV) were operated by four participating partners in a side-by-side measurement setup to observe the same emission sources under similar conditions. A novel laser spectrometer with significantly better precision specifications as compared to the other instruments was used for

the first time for emission monitoring regarding MARPOL Annex VI regulations.

The comparison took place in the Northern European Sulphur Emission Control Area (SECA) where the allowed Fuel Sulphur Content (FSC) is limited to $0.10\%S_{m/m}$. In total, 966 plumes that originated from 436 different vessels were analysed in this study. At the same time, fuel samples, obtained from 34 different vessels, and bunker delivery notes (BDN) from five frequently monitored vessels were used as references to assess the uncertainties of the different systems. Most measurement

systems tended to underestimate the FSC found from fuel samples and BDNs. A possible relation was seen to high relative humidities above 80%. The lowest systematic deviations were observed for the airborne systems and the novel laser spectrometer. The latter showed the lowest total uncertainty of $0.05\%S_{m/m}$ (confidence level: 95%) compared to other stationary sniffer systems whose total uncertainties range from 0.08 to $0.09\%S_{m/m}$. The two UAV-borne systems showed total uncertainties of 0.07 and $0.09\%S_{m/m}$, respectively. Overall, it was found that non-compliant vessels with an actual FSC of the

combusted fuel above 0.15 to $0.19\%S_{m/m}$ can be detected by the compared systems with 95% confidence.





## 1 Introduction

The International Maritime Organisation (IMO) recognized the impact of shipping emissions to the atmosphere on health, environment, and climate. In 1997, the IMO amended its convention for the prevention of pollution from ships (MARPOL) by introducing measures to gradually decrease the emissions of air pollutants, including sulphur oxides ($SO_x$), nitrogen oxides

($NO_x$), and particulate matter.

The emission of $SO_x$ is directly related to the sulphur content in the fuel which is oxidized during the engine combustion process and is emitted to the air as part of the exhaust gas. To limit the emission of $SO_x$, the IMO defined upper limits for the content of sulphur in the fuel as defined in MARPOL Annex VI, regulation 14. Alternatively, ships are allowed to bunker fuels with higher fuel sulphur contents (FSC) if they are equipped with exhaust gas cleaning systems to remove sulphur compounds.

Such systems are commonly known as *scrubbers*. The allowed emission of $SO_x$ when using scrubbers must not exceed the equivalent levels of compliant fuels. The regulation was first implemented in 2005. Since then, the global FSC cap was gradually reduced from $4.50\% S_{m/m}$, i.e. percent mass of sulphur per mass of fuel, to $0.50\% S_{m/m}$ in 2020. The European Union adopted the IMO regulation in its EU-directives 1999/32/EC and 2012/33/EC (EU, 1999, 2012). The regulation also established special Sulphur Emission Control Areas (SECA) with more stringent rules on FSC. This includes the Baltic Sea

and the Norths Sea including the English Channel, which became fully implemented as SECAs in 2006 and 2007, respectively. The level of the maximum allowed FSC has been stepwise reduced until 2015 and is currently limited to $0.10\% S_{m/m}$.

EU directive 2015/253/EC requires that at least 10% of the total number of individual vessels that are calling the relevant member state per year need to be inspected on-board which includes taking fuel samples. The total number of vessels is derived from the average number of ships of the three preceding years (European Commission, 2015). It depends on the location of

the called state what fraction of the fuel samples need to be analysed for sulphur. If the state is outside SECA 20% of the samples need to be analysed, 30% if it is partly within, and 40% if the state is entirely within the SECA. The number can be reduced by 50% if remote sensing methods are used to monitor the FSC of individual vessels.

The compliance of vessels with regulated sulphur requirements can be remotely assessed by analysis of the chemical composition of the emitted exhaust gases (Mellqvist and Berg, 2010). As in detail described in section 2.1, the measured ratio

of sulphur dioxide ($SO_2$) to carbon dioxide ($CO_2$) in the emitted plume is directly linked to the FSC. The analysed plumes can be allocated to individual nearby vessels using simultaneously measured data on wind direction and speed in combination with the identity, location and speed data received with the Automatic Identification System (AIS) information transmitted by each vessel.

Meanwhile several countries have implemented their own remote compliance monitoring strategies. They are mostly based on

in-situ systems that extract air samples of the ship exhaust plumes, which are probed using gas analysers. The volume mixing ratio (VMR) of pollutants changes as the exhaust plume reaches the monitoring station. The contribution of the vessels exhaust gas to the pollutant's concentration is estimated by comparing the respective VMRs in the exhaust plume to the background levels of the ambient air when the exhaust plume is not present. The in-situ systems are commonly referred to as *sniffers* and



are used around the world in different configurations (Beecken et al., 2019). Several states such as Germany, the Netherlands, and Sweden currently apply on-shore stationary systems near shipping lanes at rivers, harbour entrances or bridges (Alföldy et al., 2013; Balzani Lööv et al., 2014; Kattner et al., 2015; Mellqvist et al., 2017b). Similarly, mobile platforms equipped with sniffers, such as patrol vessels can be used to monitor by-passing vessels (Beecken et al., 2014b). Stationary systems are usually applied to monitor vessels near shore and are fully automized to run continuously. Sniffer systems can also be used to actively trace the exhaust plumes using airborne platforms such as drones (Explicit, 2016), helicopters (Explicit, 2018), or manned aircraft (Berg, 2011; Beecken et al., 2014a; Mellqvist et al., 2017b, a; Van Roy et al., 2022a, b). Airborne measurements allow for monitoring of vessels at any reachable position in the open sea, whereas UAV systems can also be employed for near shore monitoring.

The project SCIPPER: Shipping Contributions to Inland Pollution Push for the Enforcement of Regulations was started in 2019 with one of the objectives being to provide evidence on the operational performance and capacity of different techniques for ship emissions monitoring to contribute to the enforcement of the relevant regulations.

This study focuses on the performance of different sniffer systems in characterising their reliability to remotely monitor vessels' compliance to sulphur limits. Within SCIPPER, an elaborate measurement campaign took place near the river Elbe in Germany, about 10 km downstream of the port of Hamburg. During this campaign, five state-of-the-art sniffer systems monitoring highly diluted ship emissions at distances of several hundreds of meters are compared to a novel system using laser-spectroscopy of high sensitivity and to compact-sized, UAV-borne mini-sniffer systems that typically sample much less diluted ship plumes at a range of around 50 m from the ship funnel. The results of the measurements with these systems are compared to analysed fuel samples and bunker delivery notes of monitored vessels.

## 2 Methods

Eight different sniffer systems for the remote monitoring of ship emissions have been compared in the field between 7 September and 15 October 2020 at the Elbe River, at about 10 km downstream of the port of Hamburg. The tested systems are normally deployed by the participating groups in different locations supporting local authorities to target suspiciously operating vessels with respect to MARPOL Annex VI. For this study, these systems were benchmarked by sampling of the same vessels in the same location. The remotely assessed FSCs were further compared to the analysis of fuel samples taken as reference, onboard of selected, measured vessels or to bunker delivery notes that were provided by shipping companies.

The participating groups that provided their instrumentation for this intercomparison study were Chalmers University of Technology from Sweden, Explicit ApS from Denmark, the German Maritime and Hydrographic Agency (BSH) and the Netherlands Organization for Applied Scientific Research (TNO).





## 2.1 Remote assessment of fuel sulphur content

The FSC is expressed as percent of sulphur mass over the mass of fuel in the unit $\%S_{m/m}$. Using remote measurements, the
FSC is calculated according to Eq. 1 (Balzani Lööv et al., 2014):

$$FSC_{S\%_{m/m}} = \frac{M(S) \cdot \int [SO_{2,plume} - SO_{2,background}]_{ppb} \, dt}{10 \cdot M(C)/0.87 \cdot \int [CO_{2,plume} - CO_{2,background}]_{ppm} \, dt} = 0.232 \cdot \frac{\int [\Delta SO_2]_{ppb} \, dt}{\int [\Delta CO_2]_{ppm} \, dt}, \quad (1)$$

where *[ΔSO₂]* and *[ΔCO₂]* are the VMRs above their respective background levels, respectively for $SO_2$ and $CO_2$, in the ship
exhaust plume. The VMR of $SO_2$ and $CO_2$ are each integrated in time for the whole plume. With that, differences in instrument
response times of the individual gas analysers are compensated for. The factor 0.232 relates to the molecular masses *M(S)* and
*M(C)* of sulphur and carbon, respectively as well as an assumed carbon content in fuel of $87\%_{m/m}$ (MEPC/Circ. 471, 2005). A
conversion factor of 10 is also considered to express the FSC as $\%S_{m/m}$ when *[ΔSO₂]* is expressed in parts per billion (ppb) and
*[ΔCO₂]* is expressed in parts per million (ppm).

For this calculation it is assumed that all sulphur in the fuel is emitted as $SO_2$ after combustion and that other emitted sulphur
species can be neglected. Grigoriadis et al., 2021, showed that sulphur to sulphate conversion is not more than 0.8% for
distillate fuels, with an expected value of around 0.5% for the slow cruising loads at the Elbe River. This is the maximum bias
expected from such an approximation. Likewise, it is assumed that carbon in the fuel is nearly completely converted to $CO_2$
in the combustion process (Moldanova et al., 2009). Hence, with measured VMR of $CO_2$, the emitted $SO_2$ can be directly
related to the amount of fuel being used.

Some groups also reported negative results for the estimated FSC. This could happen after baseline correction for very low
$SO_2$ signals or an overcompensation of the NO cross-sensitivity by the $SO_2$ instrument. Any estimated negative FSCs were set
to zero for the comparison.

## 2.2 Measurement systems

All eight systems that are compared in this study estimate the FSC of passing vessels by analysing air samples from the emitted
exhaust plumes and are therefore denoted as sniffers. In this study six shore-based stationary sniffers and two compact UAV-
borne mini sniffers were compared to each other.

The systems are individually short-named by the team that operates them, i.e. BSH (bsh), Chalmers (cha), TNO (tno), and
Explicit (exp). Additionally, the systems are distinguished by their type, i.e. standard sniffer (std), laser spectrometer (las), or
unmanned aerial system (uas). The three systems operated by BSH are all standard sniffers. Therefore, these are further denoted
by their make, i.e. Airpointer (ap), and Horiba (hor), or their deployment type as for the mobile measurement system (mms).



### 2.2.1 Stationary systems

Stationary monitoring systems near the waterways collect and analyse, sniff, the exhaust plumes that are transported from passing-by vessels to the sniffers by wind. Their locations are generally selected to suit the local prevailing wind conditions to increase the chances of measuring ship plumes. The operational distance to the vessel is usually several hundred meters. Therefore, the sulphur dioxide detection limit of these systems needs to be able to detect the comparably low VMR differences with respect to the background levels, which are commonly in the range of only a few ppb of $SO_2$ and a few ppm of $CO_2$, and sensitive enough to capture the variance in the highly diluted plumes.

An overview of the specifications of the instrumentation used by the individual stationary systems is provided in Table 1.

**Table 1: Instrumentation specifications of stationary systems. The systems bsh.ap and bsh.mms use the same instrumentation but are individual systems on different physical platforms.**

|  | BSH |  | TNO | Chalmers |  |
|---|---|---|---|---|---|
| System name | bsh.ap[p] bsh.mms | bsh.hor[p] | tno.std | cha.std | cha.las |
| data collection period | 7 Sep. – 15 Oct. | 7 Sep. – 15 Oct. | 8 Sep. – 15 Oct. | 20 Sep. – 14 Oct. | 20 Sep. – 14 Oct. |
| System make | mlu-recordum | Horiba | in-house | in-house | Aerodyne |
| $SO_2$ | Airpointer | Horiba APSA-370 | Thermo 43i-TLE | Thermo 43i-TLE | TILDAS Dual Laser Trace Gas Analyzer |
| *measurement principle* | *UV-fluorescence* | *UV-fluorescence* | *UV-fluorescence* | *UV-fluorescence* | *TILDAS[T]* |
| *precision [ppb]* | *1% of reading, but at least 1 ppb* | *0.5% of reading, but at least 0.5 ppb* | *1* | *1* | *0.015* |
| *detection limit [ppb]* | *2* | *0.5* | *3* | *3* | *0.06* |
| *response time (T90) [s]* | *40* | *40* | *40* | *40* | *1* |
| *sampling rate [Hz]* | *0.1* | *0.2* | *1* | *1* | *1* |
| *cross-sensitivity to NO [%]* | *0.7 to 1.5* | *0.5 to 1.0* | *0.8* | *1.5* | *no cross-sensitivity* |
| $CO_2$ | LI-COR 840A | LI-COR 840A | LI-COR 7000 | Picarro G2301-m | TILDAS Dual Laser Trace Gas Analyzer |
| *measurement principle* | *NDIR[N]* | *NDIR[N]* | *NDIR[N]* | *CRDS[C]* | *TILDAS[T]* |
| *precision [ppm]* | *< 1* | *< 1* | *< 1* | *0.06* | *0.06* |
| *detection limit [ppm]* | *0.2* | *0.2* | *0.2* | *0.2* | *0.2* |
| *response time (T90) [s]* | *1* | *1* | *1* | *1* | *1* |
| *sampling rate [Hz]* | *0.1* | *0.2* | *1* | *1* | *1* |
| $NO, NO_2, NO_x$ | Airpointer | Horiba APNA-370 | EcoPhysics CLD700-AL | Thermo 42i-TL | (not applicable) |
| *measurement principle* | *chemi-fluorescence* | *chemi-fluorescence* | *chemi-fluorescence* | *chemi-fluorescence* | *~* |
| *precision [ppb]* | *1% of reading, but at least 1 ppb* | *0.5% of reading, but at least 0.5 ppb* | *0.5* | *0.3* | *~* |
| *detection limit [ppb]* | *1* | *1* | *1* | *1* | *~* |
| *response time (T90) [s]* | *30* | *30* | *2* | *1* | *~* |





| sampling rate [Hz] | 0.1 | 0.2 | 1 | 1 | ~ |
|---|---|---|---|---|---|

[p] The systems bsh.ap and bsh.hor are permanently operated at this site.

[N] NDIR: NonDispersive InfraRed spectroscopy

[C] CRDS: Cavity Ring-Down Spectroscopy

[T] TILDAS: Tunable Infrared Laser Direct Absorption Spectroscopy


During this campaign a novel, $SO_2$ and $CO_2$ analyser based on Tunable Infrared Laser Direct Absorption Spectroscopy (TILDAS) with a particularly high sensitivity on $SO_2$ operated by Chalmers was used for the first time for compliance monitoring. Its detection limit is about 10 to 50 times below that of the state-of-the-art stationary instruments. It is also characterised by its fast response time of 1 s and $SO_2$ and $CO_2$ are analysed synchronously. The other stationary systems do

not significantly differ from each other in their underlying measurement principles. The stationary instruments are henceforth distinguished into two groups, standard sniffers composed of the bsh.ap, bsh.hor, bsh.mms, cha.std, and tno.std systems and the highly-senstitive one with the cha.las system.

### 2.2.2 Airborne mini sniffer systems

Airborne mini sniffer systems are flown directly into the vessel's plume. Hence, plume samples can be collected much closer

to the funnel exit, with much less dilution as compared to stationary systems. Typical sampling distances for these systems are in the range of 50 to 100 m from the funnel's exit and the UAVs are piloted into sweet spots within the plume with a target for $CO_2$ VMR being 100 to 200 ppm above the background. Carbon dioxide is measured using a compact Non-Dispersive InfraRed (NDIR) sensor while other species such as $SO_2$, NO, and $NO_2$ are measured using Electro-Chemical (EC) sensors. Typical VMRs in the sweet spots are in the range of a few tens of ppb for $SO_2$, depending on the vessels' fuel and the presence of any

abatement systems, and in the order of single digit ppm for NO and several hundred ppb for $NO_2$, respectively.

Two mini sniffer systems onboard drones were employed in this study. A commercial UAS by Explicit, in this study named as exp.uas. This kind of sensor is used for emission monitoring on a regular basis (Explicit, 2016, 2018). A second UAS, applied by Chalmers, was used for the first time as an experimental system, herein named as cha.uas. Both drones were deployed between 13 and 16 September 2020.

The specifications of both systems are similar. The compact NDIR sensors provide precision and detection limits below 10 ppm for $CO_2$ and a $T_{90}$ response time of about 20 s. The precision of the $SO_2$ sensors is around 7 ppb, and their detection limit is around 20 ppb with a $T_{90}$ response time of 20 s. The $SO_2$ sensors show a strong negative response to $NO_2$ by 120% which is corrected for based on simultaneous $NO_2$ measurements. The NO sensors in both UAS have a precision and detection limit below 40 ppb and a $T_{90}$ response time of about 25 s. The $NO_2$ sensors have a precision and detection limit below 20 ppb

and a $T_{90}$ response time of below 80 s.

It was observed that the response times for the EC sensors depend on the actual VMR and are faster and in the order of a few seconds to about 20 s for the typically observed VMR ranges during ship emission monitoring operations mentioned above. The sampling rate is 1 Hz for both, exp.uas and cha.uas.



## 2.3 Calibration

The calibration of each stationary system was conducted in a similar way by the different groups. Dry zero gas, which is clean of any of the targeted species and dry calibration gas at known VMR levels were successively fed to the instruments for a certain amount of time until the instruments response has stabilized. However, the calibration schemes and applied gas VMRs for the different systems differed depending on the group, see Table 2.

**Table 2: Calibration parameters for autonomously operated stationary systems by group.**

|  | BSH | TNO | Chalmers |
|---|---|---|---|
| typical calibration interval | 6 months* | 1 month | 10 to 20 days |
| calibration interval during campaign | begin and end of campaign | begin and end of campaign | daily |
| mixing rations of calibration gases |  |  |  |
| $SO_2$ [ppb] | _100_ | _100 to 200_ | _330_ |
| $CO_2$ [ppm] | _300; 900_ | _450_ | _300_ |
| NO [ppb] | _200; 100 to 500**_ | _400***_ | _300_ |

\* Additional regular automatic validation with internal VMR standards supplied by permeation tubes at a 25-hour interval.

\** Adjustable dilution with a gas mixing chamber from 40 ppm to 5 VMR levels between 100 and 500 ppb. Used for cross-sensitivity evaluation.

\*** A Sonimix 6000 C2 dilution system was used the vary the NO concentration during calibration between 0 and 90 ppb

The exp.uas was found to be stable and reliable for more than 100 h of operation through drift and performance tests conducted by a reference laboratory according to ISO-standard EN ISO 6145-1. New sensors are calibrated before in-field deployment by a reference laboratory to ensure that the units work within the given uncertainties. These units are replaced after 100 hours of operation or latest one year after their production.

Chalmers calibrated its mini sniffer system before the campaign against reference analysers in the laboratory by simultaneously
exposing the systems to gas mixtures of different VMRs.

## 2.4 Uncertainty

The groups used individual approaches to assess the uncertainty of their measurements and which they reported along with the estimated FSC. For the outcome of this study, the results are shown in a harmonized representation based on an expended uncertainty developed from the intercomparison to collected reference data.

### 2.4.1 Reported uncertainty

So far, the different measurement groups used their own calculation and reporting processes, which are presented in detail in Beecken et al., 2019, and Mellqvist et al., 2022. Hence, any reported uncertainties in the FSCs are based on uncertainty calculation procedures that may differ between the various teams. In the current study, two alternative approaches were followed to determine the uncertainty in the measurements of each system.





In the first approach (type A), the uncertainty is estimated only based on the characteristics of the measurement of each plume. It is based on an error budget which generally includes the standard deviations of the individual signals of the target species, i.e. $CO_2$ and $SO_2$, as well as individually observed uncertainties from the calibration and due to any cross-sensitivities. On top of this, BSH considers the uncertainty in the assumption of complete conversion of sulphur to $SO_2$ and carbon to $CO_2$ and the influence of relative humidity by estimated impact factors. Chalmers also considers differences of the background levels before

and after the detected plume, and the variability between successive calibrations.

Explicit, on the other hand, reports their uncertainties according to a pre-determined scheme (type B). The uncertainties of the sensors are characterized according to ISO 61451 and ISO/IEC Guide 983:2008 at a reference laboratory and are validated by comparisons to fuel samples during field measurements. The tests were conducted under different representative environmental conditions and different mixing gas ratios, corresponding to different FSCs and distances to the emission stack. Also, sensor

cross-sensitivities were characterized in this manner. It was found that the observed uncertainties only showed a significant dependence on the calculated $SO_2$-to-$CO_2$ ratio, hence FSC. Therefore, the reported uncertainty for exp.uas is only a function of the calculated FSC of the measured plumes.

Currently there also is a difference between the confidence levels of the reported uncertainties, which are also used to report results to authorities. While Chalmers reports uncertainties at a confidence level of 95%, the other groups report the uncertainty

as one standard deviation which corresponds to a confidence level of about 68%. The analysis is intentionally using the reported values to show the potential and need for harmonization.

**2.4.2 Expanded uncertainty**

The method of expanded uncertainty is used to describe the performance of the systems based on the results from the comparison of the estimated FSC from the plume measurements to the expected FSC. It is expressed according to

ISO/IEC 98-3:2008, and Magnusson and Ellison (2008). The total uncertainty can be calculated by

$$U_{total} = k \sqrt{\left( U_{random}^2 + \left( \frac{\overline{U}_{bias}}{\sqrt{3}} \right)^2 \right)} \tag{2}$$

where, $U_{random}$ describes the random uncertainty which corresponds to the standard deviation found from the comparison. The contribution of the bias to the total uncertainty is calculated based on an assumed rectangular distribution, as a conservative estimate of the probability distribution function for the bias. Hence, the observed mean deviation in the bias term, $\overline{U}_{bias}$, is

accordingly divided by $\sqrt{3}$ for the calculation of its standard deviation. The overall distribution of the total uncertainty is assumed to follow a t-distribution. The factor $k$ depends on the confidence level, which in this study is chosen to be 95%, and the number of observations. In this study $k$ was found to be in the range between 1.99 to 2.07, depending on the total number of comparisons of the respective system.





## 2.5 Location

This SCIPPER campaign took place in Wedel near Hamburg at the Elbe River waterway connecting the port of Hamburg with

the North Sea. According to IMO's regulation the maximum FSC that seagoing ships are allowed to use here is $0.10\%S_{m/m}$.

The measurement site at 53.5696°N and 9.6917°E has been in use by BSH for ship emission monitoring since September 2014

with about 40'000 vessels passing by that site annually. During the campaign, ships passed-by with an average speed over

ground of $10.5 \pm 2.7$ kn. The site is located at the northern banks of the river, shown in Figure 1, considering the predominant

wind from south-west. The distance from the stationary measurement systems and the launch site of the exp.uas system to the

shipping lane was approximately 500 m. The cha.uas system was launched near the river about 2.7 km north-west of the main

campaign site to avoid interferences between the two drone operations.

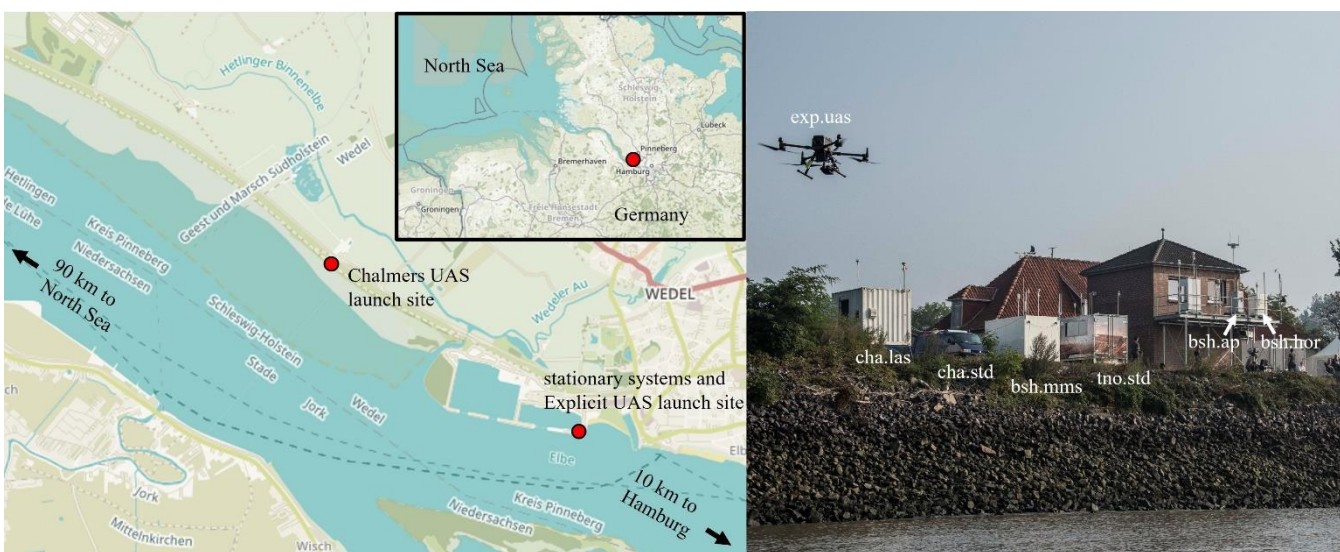

**Figure 1: Left panel: location of the measurement site at the northern banks of the Elbe River in Wedel near Hamburg
(© OpenStreetMap contributors 2023. Distributed under the Open Data Commons Open Database License (ODbL) v1.0.). Right
panel: picture of the positioning of the stationary systems as seen from river (© BSH).**

BSH permanently operates two stationary standard sniffer systems on this site, bsh.ap and bsh.hor. The other systems were

located within a 20 m perimeter to the west of BSH's permanent installation, unobstructed along the waterline with similar

distance to passing vessels. The inlet height of each system was between 7 and 8 m above mean water level. This varied during

the measurements with the local tidal range of about 3 m.

## 2.6 Fuel samples and bunker delivery notes

The measurements were complemented by fuel samples obtained by the Hamburg Water Police from ships that had passed the

measurement site when they were at berth in the Hamburg port area. The fuel samples that were used for comparison were all





taken from the fuel lines to the main engine. They were analysed in BSH's own ISO 17025 certified laboratory for their sulphur content. In total, 29 fuel samples could be related to plumes measured by at least one of the systems.

Some vessels were frequently observed during the measurement period, but a fuel sample could not be obtained from them. Instead, the shipping company of five frequently measured dredger vessels voluntarily shared all relevant bunker delivery
notes (BDN) that were used as basis for comparison.

The measured FSC from the fuel samples and the FSC data retrieved from the BDNs assumed to be representative with respect to the true FSC at the time the remote measurement took place and are used as the expected FSC in the comparison.

## 3 Results and discussion

### 3.1 Comparison

The overall campaign period lasted from 7 September to 15 October 2020. As presented in Table 1, the autonomous stationary systems nearly continuously measured throughout this period. The UAV-borne mini-sniffers that were actively piloted into the plumes were operated for four consecutive days between 13 and 16 September.

Altogether 966 plumes from 436 unique vessels were measured. Of these, both UAVs captured 70 individual plumes from 58 different vessels. The different systems did not always capture the same plumes, but 724 individual cases were measured by
at least two systems. However, only a few plumes were simultaneously measured by at least one stationary and at least one UAV-borne system due to unfavourable wind conditions that impacted the transport of the exhaust plumes to the shore-based stationary systems at the time of the UAV flights. Nevertheless, due to the available fuel samples and frequently passed vessels whose exhaust was captured on many occasions by the different systems, a comparison was still feasible.

An intercomparison of the calibration standards by TNO's laboratory showed a deviation of up to 40% from the manufacturers'
specifications. In this case, the deviation exceeded the specified uncertainty significantly. This highlights the need to validate VMRs of the calibration gases. A correction to the affected data was applied for this study.

For benchmarking all systems on the same basis, the estimated FSCs for individual plume measurements of the systems were compared to corresponding FSCs measured from fuel sampling or retrieved from the BDNs. For the fuel samples, the remote measurements of each vessel throughout the entire campaign period were compared to the FSC of the fuel sample to increase
the number of comparisons, under the assumption that the true FSC of the measured vessels did not change significantly within the observation period. For the comparison of the estimated FSCs to FSC data from BDNs, the reported FSC data of the most recent BDN for each specific vessel was considered. In total, 145 individual plumes that were measured by at least one of the systems corresponded to measured FSCs from fuel sampling or BDNs.

The results of the FSCs from the fuel sample analyses and the BDNs were not made available to the participating teams to
keep the comparison unbiased in form of a blind comparison.

In Figure 2 the absolute deviation of the individual plume observations from the fuel sample results is presented as a function of the reported uncertainty by each system. This figure helps summarizing how well the estimated FSC from each system





matches with the expected FSC from the fuel sampling considering the individually reported uncertainties. Three distinct cases can be identified. Estimates that lie in the unshaded area match with the expected FSC of the respective vessel within the range of the individually reported uncertainty. Those estimates that lie in the upper grey shaded area are higher than the measured FSC of the fuel sample plus the reported uncertainty and correspond to overestimates with respect to the expected FSC. In the opposite case, values in the lower shaded area underestimate the expected FSC.

The high frequency of data points in the lower shaded area shows that stationary systems mostly seem to underestimate the FSC beyond uncertainty. Even if one would neglect the bias, the reported uncertainty for the standard sniffer systems (bsh.hor, bsh.ap, bsh.mms, cha.std, and tno.std) appears to be too small when compared to the spread of the estimations for each individual system along the y-axis. This is particularly apparent for reported uncertainties below $0.02\,\%\,S_{m/m}$. The distribution of the reported uncertainties also reflects the differences in the chosen confidence levels between the different groups. BSH, Explicit, and TNO reported their uncertainties as a single standard deviation corresponding to a confidence level of about 68% while Chalmers reported their results with 95% confidence. This becomes particularly obvious for the cha.std and cha.uas estimates which spread over a wider range along the x-axis compared to other systems. However, apart from the apparent bias, the absolute deviation of the novel laser-spectrometer, cha.las, exhibit a comparatively small spread, reflecting the higher precision of this system as shown in Table 1. And unlike the other compared systems, cha.las shows no strong cross-sensitivity to other gases, e.g. NO.



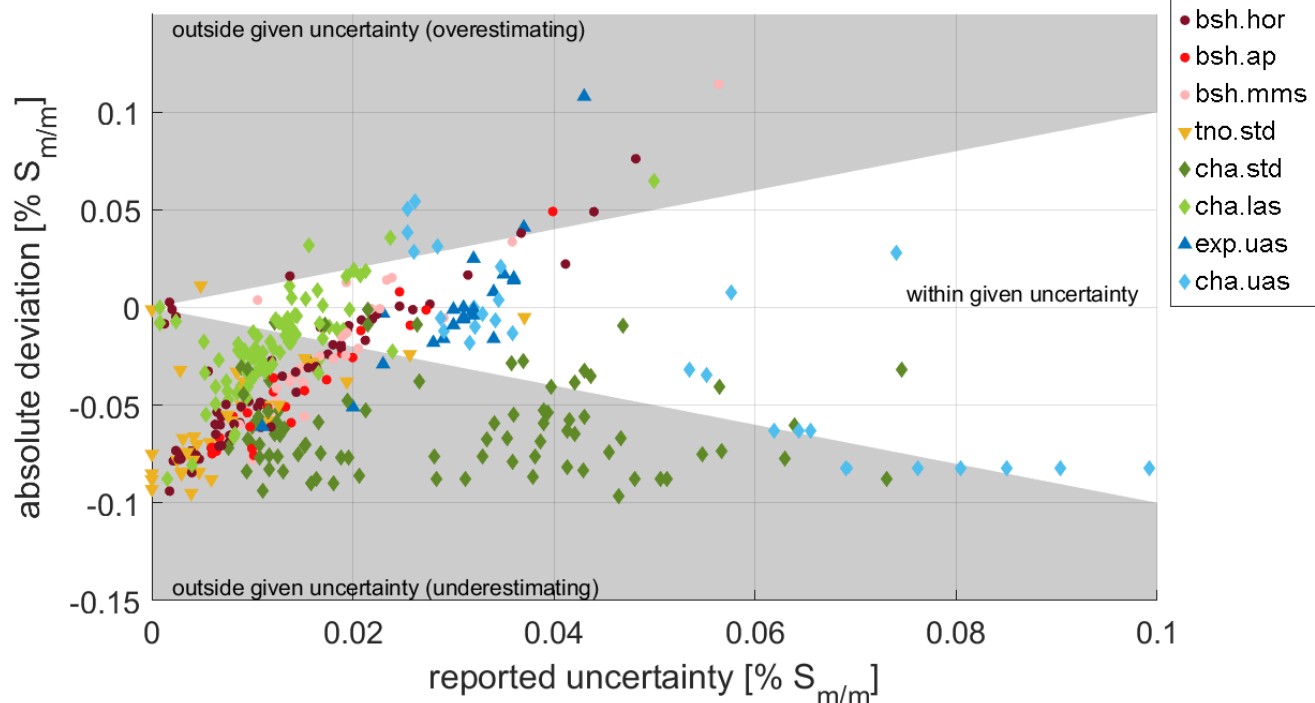


**Figure 2: Comparison of individual estimates per system to the expected FSC from fuel sampling and BDN as a function over of the reported uncertainty. The shade of the areas is indicative of whether a measurement matches (unshaded area), overestimates (upper grey-shaded area), or underestimtates (lower grey-shaded area) the expected FSC.**

The exp.uas system correctly estimated the FSC in 78% of the cases with an uncertainty corresponding to a confidence level of about 68%. There is no apparent tendency for exp.uas to overestimate nor underestimate. With the cha.uas system the FSC was correctly quantified in 68% of the cases using a broader confidence interval of 95% with a tendency towards lower values for the rest. All stationary systems appeared to experience a systematic negative bias and matched within uncertainty ranges only within 6 to 41% of all cases. For these systems, there are only a few cases of overestimation while underestimation is

obvious in 56 to 91% of the cases.

Figure 3 summarizes the deviations of the estimates per system to the fuel sample and BDN results. The mean values of the underestimation of the systems operated by BSH are ranging between -0.017 and -0.040%$S_{m/m}$. This deviation is -0.057 and -0.062%$S_{m/m}$ for the tno.std and cha.std system, respectively. For the highly sensitive cha.las it is 0.020%$S_{m/m}$. While there is no systematic deviation for exp.uas, this number is 0.020%$S_{m/m}$ for cha.uas. A potential cause for the underestimation of the

FSC by most systems is discussed in section 3.2.





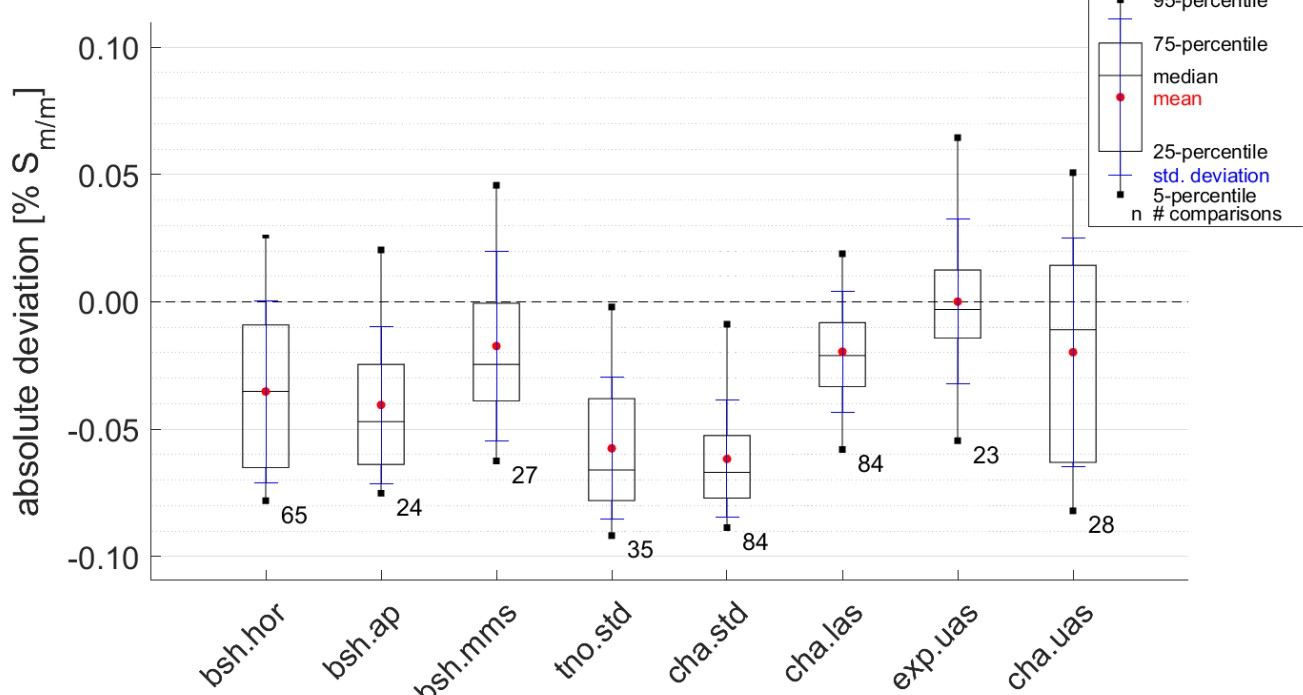

**Figure 3: Comparison of the deviations per system from the expected FSC determined by fuel sampling or BDN. The number next to each bar corresponds to the number of observations.**


The benchmarking of the systems based on the comparison to the fuel samples is shown in Table 3. Grouping the results into the underlying technologies gives the following results. The total uncertainties of the standard sniffers, which are calculated according to Eq. (2), are in a range between 0.079 and $0.088\%\,S_{m/m}$. The cha.las shows a total uncertainty of $0.052\%\,S_{m/m}$. The lower uncertainty of the cha.las compared to the standard sniffers reflects the higher precision especially towards $SO_2$ at VMRs

which are typically in the order of only a few ppb for stationary systems given the typical range of distances of several hundred metres to the vessels. These VMRs are close to the detection limits of the $SO_2$ monitors in the standard sniffers but significantly higher than the detection limit of 0.06 ppb for cha.las, see Table 1.

While the monitors in all stationary systems are made for analysing pollutants at trace gas levels, the mini sniffer systems aim for higher sample gas VMRs. The total uncertainty of exp.uas of $0.067\%\,S_{m/m}$ is between those of cha.las and the stationary

systems, while cha.uas showed a significantly higher total uncertainty of $0.095\%\,S_{m/m}$. The deviation in the total uncertainty between the two UAV-borne mini sniffers could be related to a reportedly longer distance to the emission source of 100 to 200 m for cha.uas compared to around 50 m for exp.uas. At such distances the plume is more diluted consequently the VMR ranges are much lower leading to a large uncertainty. Moreover, the quality of the sample collection, i.e. the pilots capability to find and remain at the sweet spot within the plume, is another factor influencing the uncertainty. Considering the response





characteristics of the sensors, a certain residence time at sufficiently stable mixing ratios in the plume improves the quality of the measurements.

**Table 3: Re-evaluated uncertainties for the systems based on the fuel sample comparison. The presented random and total uncertainties correspond to a 95% confidence level assuming a normal distribution.**

| system | bias<br>$\bar{U}_{bias}$<br>[$\%S_{m/m}$] | random uncertainty<br>$k \cdot U_{random}$<br>[$\%S_{m/m}$] | total uncertainty<br>$U_{total}$<br>[$\%S_{m/m}$] |
|---|---|---|---|
| bsh.hor | -0.035 | 0.071 | 0.082 |
| bsh.ap | -0.040 | 0.064 | 0.080 |
| bsh.mms | -0.017 | 0.076 | 0.079 |
| tno.std | -0.057 | 0.056 | 0.088 |
| cha.std | -0.062 | 0.046 | 0.084 |
| cha.las | -0.020 | 0.047 | 0.052 |
| exp.uas | 0.000 | 0.067 | 0.067 |
| cha.uas | -0.020 | 0.092 | 0.095 |


### 3.2 Possible causes for the negative bias in FSC estimates

As gaseous $SO_2$ dissolves in water (Terraglio and Manganelli, 1967), absorption of $SO_2$ on wet surfaces in the inlet section of the systems is seen as one possible cause of the prominent negative bias, most prevalent in case of the stationary systems. For this comparison, continuously recorded meteorological data at the location of bsh.mms was used. Over the time of the

campaign the relative humidity varied between 50 and 100%. The absolute deviation of individual estimates to the expected values from the fuel sampling and BDNs are plotted in relation to the relative humidity present at the time of the plume measurements in Figure 4. Most measurements were conducted while the relative humidity was above 70%, hence the statistically relevant range is limited. For most systems, i.e. bsh.hor, cha.std, cha.las, and tno.std, a relation between deviation and relative humidity was observed. The results show that such an effect is increasingly prominent when the relative humidity

exceeds 80%. However, these results can only be considered suggestive of associations and other meteorological and operational parameters are needed for a proper analysis and potential correction. The actual effect might differ between the instruments. It can be expected that the surface area, flow characteristics, and residence time of sampling lines could be important parameters. Also, the dew point in the inlet section is not only dependent on the relative humidity but also on other parameters such as air pressure in the sampling line and the temperature along its surfaces.

Separate testing of each individual system under controlled laboratory conditions would be required to describe the influence of these parameters and to develop an algorithm to correct measured values according to humidity. This hypothesis is strengthened by the observation of a significantly lower to even no apparent bias of the exp.uas, where there is only little surface area and low residence times due to short tube lengths in the order of a few decimetres, and no filter presence, where





condensation could take place. For example, in the case of the stronger affected cha.std and cha.las the inlet tubes were already
exceeding a length of 3 m. On the other hand, heating the inlet section might prevent condensation and make a mathematical
correction unnecessary.

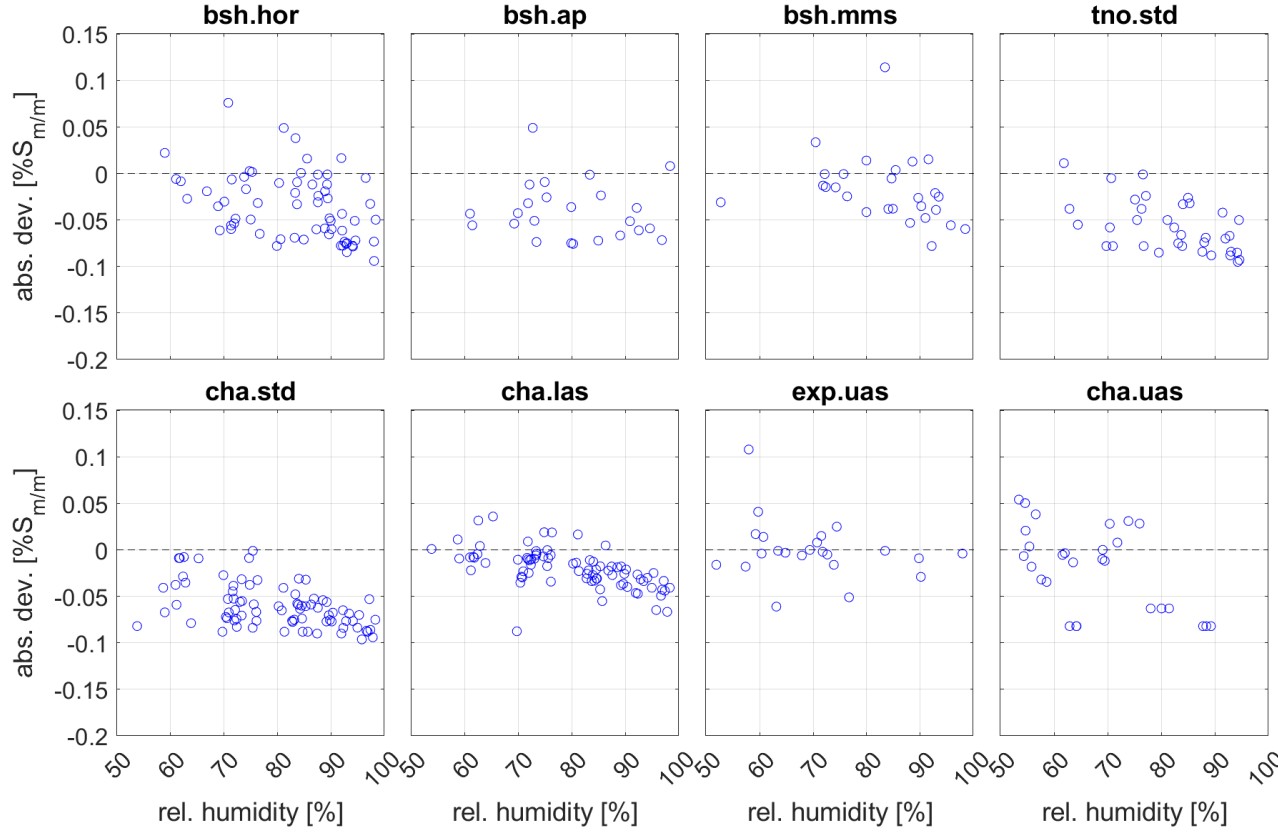

**Figure** 4**: Dependence of absolute deviation of each system from the expected FSC on relative humidity at the time of**
**the measurement.**

The bias could also be influenced by a wrong estimation of cross-sensitivities. The UV-fluorescence instrument for the analysis
of $SO_2$ in the standard sniffers is sensitive also to NO and Volatile Organic Compounds (VOC). These instruments were tested
for the influence of NO and the correction factors applied accordingly to minimize this effect. Only the cha.std system was set
to measure $NO_x$ instead of NO during most of the time of the campaign period. In this case a constant fraction of NO in NOx
was assumed and used for correction. This causes some additional uncertainty as this fraction varies with plume age depending
on the ambient conditions.



# 4 Conclusion

Different state-of-the-art and novel instruments were assessed as measurement systems to remotely measure the FSC of individual vessels by analysing the emitted exhaust in a side-by-side study. The measurements were carried out as part of a campaign conducted in Germany along the Elbe River in 2020 in the framework of the European Commission project SCIPPER.

The compared systems were grouped into three different classes. First, five stationary sniffer systems which are using similar measurement principles, i.e. UV-fluorescence and in most cases NDIR for quantification of $SO_2$ and $CO_2$, respectively, are regarded as one class. Second, a novel, highly sensitive system based on laser spectroscopy has been used for the first time in the field of ship emissions monitoring. Finally, two UAV-borne mini sniffer systems were employed, and these were actively piloted into the plume closer to the ships funnel where the concentration of species is higher than in the case of the remote stationary systems.

In total, 966 individual exhaust plumes have been analysed on different occasions. The measured ships were sailing in SECA when they were measured. They were hence obliged to use fuel that contains $0.10\%\,S_{m/m}$ or less or alternatively run scrubber systems to limit their sulphur content in the exhaust. So, the performance of the instrumentation is assessed considering the lowest IMO limit which is currently in place worldwide.

Measured FSC from fuel samples taken by the Water Police at the port of Hamburg during the campaign period and FSC data retrieved from BDNs were used as a reference to evaluate the absolute deviations of the results from the monitoring to the actual FSC of the fuels used. All systems, except for the UAV-borne exp.uas underestimated the evaluated reference in most cases. On average, the standard sniffers underestimated the references by a mean of 0.02 to $0.07\%\,S_{m/m}$, the cha.las by $0.02\%\,S_{m/m}$ and the cha.uas by $0.02\%\,S_{m/m}$ while the average deviation of the exp.uas was zero.

The reported uncertainties based on each groups' own uncertainty estimation did underestimate the deviations from the fuel samples, except for the mini sniffer systems. Considering the reported uncertainties, the standard and high-sensitive sniffer systems do match the analysed FSC from the fuel sample in 6 to 41% of the cases. With a tendency for underestimating the FSC in 56 to 91% of the cases. The drone-borne mini sniffers on the other hand did match the expected FSCs in 78% of the cases for exp.uas and in 68% of the cases for cha.uas.

High relative humidities during the measurements tend to correlate with this underestimation of the FSC for most systems. This observation might be explained by condensation of water vapour on the walls of the tube or on filter surfaces in which $SO_2$ could dissolve. These effects might be mitigated by reduced tube lengths and heating of the tubes to prevent condensation. However, further research is needed for being able to provide stronger conclusions about this humidity effect and to evaluate the significance of it compared to potential other reasons.

Total uncertainties were calculated based on the comparison to the fuel samples. These include the bias and random error corresponding to a 95% confidence level for FSCs from the fuel samples and BDNs at $0.075 \pm 0.025\%\,S_{m/m}$. For the standard sniffers the total uncertainty is in the range of 0.08 to $0.09\%\,S_{m/m}$, $0.05\%\,S_{m/m}$ for cha.las, $0.07\%\,S_{m/m}$ for exp.uas and $0.09\%\,S_{m/m}$



for cha.uas. This means that the currently applied systems are capable of reliably detecting non-compliantly operated vessels in SECAs, where the FSC is limited to $0.10\%S_{m/m}$. The certainty in the assessment is at least 95% if the observed vessel is operated with fuel that contains 0.15 to $0.19\%S_{m/m}$ or more, depending on the total uncertainties of the individual systems.

A comparison of the applied calibration standards showed deviations from the individual manufacturer certificates of the gas

VMRs in the cylinders. The actual value of the $SO_2$ VMR was 40% less than specified by the manufacturer while the manufacturer evaluated the uncertainty of the VMR in the delivered gases to be 5%. Any deviation of calibration values has a proportional effect to the observed FSC results. To maintain and assure a high measurement quality it is therefore recommended to cross-check the expected VMRs of new gas cylinders with their predecessors and to standards in reference laboratories. Also, round-robin tests between different applicating groups are deemed helpful to detect and correct for any

deviations.

Concluding our findings above, all the presented methods are suitable for remote monitoring of FSC of vessels sailing according to MARPOL Annex VI regulations. The findings of this study can be used to further improve systems and the quality assurance of procedures for emissions monitoring. Appropriate suggestions are presented in herein.

## 5 Acknowledgements

This work was conducted in the framework of SCIPPER project. The SCIPPER project has received funding from the European Union's Horizon 2020 research and innovation programme under grant agreement Nr.814893.

The authors would like to express their gratitude to the Wasserstraßen- und Schifffahrtsamt Elbe-Nordsee for providing their facilities and their highly appreciated service during the conduction of this campaign. The Water Police Hamburg is thanked for their excellent work in taking numerous fuel samples. Thanks also to the participating ship operators and responsible

personnel onboard the sampled vessels, especially Jan de Nul Group, for supporting this project by the supply of fuel samples and by providing bunker delivery notes.

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
