# Peer review of "Performance assessment of state-of-the-art and novel methods for remote compliance monitoring of sulphur emissions from shipping"

_Atmospheric Measurement Techniques, 2023_

## Author Comment (AC1)

We thank Referee 1 for the highly appreciated review. We refer to the individual comments as follows:

**General comments**

Beecken et al. describes the results of a measurement campaign undertaken to evaluate the performance of different instruments for the monitoring of pollutants emitted by ships. The manuscript focuses on $SO_x$ emissions and compares the results of the different instruments and measurement principles. The importance of proper calibration is highlighted. Generally, the manuscript is well written and fits into the scope of AMT. It can be accepted for publication after some minor clarifications.

Sect 2: Is the evaluation process the same for all instruments? Where the same evaluation programs used or where there at least common settings (for example for peak detection and ship assignment to a plume)? A brief summary of the steps of the evaluation process might be beneficial.

  AC: The principle of evaluation process is similar between the groups. However, the individual implementations are also reflecting the different instrumentations. The following sentence is added for clarification: "The exact implementations of the FSC calculation vary between the different groups according to their instrumentations and corrections for instrumental cross-sensitivities to other gas species, but all follow the principle described above. A detailed description of the instrument individual data analysis can be found in SCIPPER deliverable D2.3, Section 2 (Beecken et al., 2019). " (l. 112ff).

Sect 2.1: Here the calculation of the fuel sulphur content is explained. I feel there should be a brief description of how the baseline is defined for $SO_2$ and $CO_2$. Are the baselines defined in the same way for all instruments? Are there any additional sources next to ships that could cause enhancements in $SO_2$ and $CO_2$ at the same time?

  AC: A short description of the baseline definition is added in this section now: "The baseline which is used for subtracting the background was obtained from the ambient VMR levels before and after the plume appears in the sensor signals. " (l. 100f).
Regarding influence of other sources, a clarification is added in section 2.2.1: "The allocation of the measured plumes with individual vessels is achieved through simultaneous wind and AIS data recording. Measurements are discarded in cases where several sources cannot be distinguished, such as from potentially mixed plumes of two vessels passing the sampling site at the same time. "
(l. 135ff).
It is also added now in section 2.5 that "… there are no further sources in that area that interfere with the plume measurements. ".

Sect 2.5: Perhaps this section should be moved to the beginning of section 2, before the detailed description of the instruments, data evaluation and description of the uncertainties.

  AC: It is preferred to put this section as 2.5 because it relates and names specific systems that are first introduced in section 2.2 *Measurement Systems*. The sections 2.3 *Calibration* and 2.4 *Uncertainty* are directly connected to section 2.2 and were therefore kept in this order

Sect 3.2: Some of the instruments are specifically used by the BSH to monitor ship emissions. Was

the negative bias already known and is this usually corrected for? Is it relevant for the identification of non-compliant ships?

AC: The negative bias affected all systems except exp.uas not only BSH's systems. However, its magnitude was first noticed in this broad study due to the comparison to the high number of references, i.e. fuel samples and bunker delivery notes. It is indeed relevant because it leads to an underestimation of the actual FSC.

**Specific comments**

Line 146: How are these sweet spots identified, how long does it take to find a good position and how long does the UAV need to be in this position for an accurate measurement? Are the results significantly different for measurements outside the sweet spot?

AC: The sweet spot is described in more detail now in section 2.2.2: "Typical sampling distances for these systems are in the range of 50 to 100 m from the funnel's exit and the UAVs are piloted into sweet spots within the plume. Here, the sweet spot describes a plume region, where the expected VMRs of the species of interest can be well quantified according to the sensor specs. The guiding species is $CO_2$ with its VMR targeted to be 100 to 200 ppm above the background. Carbon dioxide is measured using a compact Non-Dispersive InfraRed (NDIR) sensor while other species such as $SO_2$, NO, and $NO_2$ are measured using Electro-Chemical (EC) sensors. Typical VMRs in the sweet spots are in the range of a few tens of ppb for $SO_2$, depending on the vessels' fuel and the presence of any abatement systems, and in the order of single digit ppm for NO and several hundred ppb for $NO_2$, respectively. Typical residence times in the sweet spots are between 30 s to several minutes. "
(l. 156ff).
The ratios do not vary significantly within the plume around this region.

Line 177: Any reason why the units are replaced after 100 hours or 1 year after their production? Is regular calibration not good enough?

AC: The EC sensors have a limited lifetime which is stress tested for the applied sensors to be at least 100 hours of operation. This is described in more detail now: "The ECse units sensors  have a lifetime of at least 100 hours of operation and can be operated for least one year after production without effects of sensor deterioration that are impacting the measurements (Explicit, 2018). They are replaced accordingly." (l. 190ff).

Line 253: The stationary instruments were set up right next to each other and should always see the same plumes, but it seems that 25% of the detected plumes were only detected by one of the instruments. Any explanations on this?

AC: To distinguish a plume from the background is challenging and the measured VMRs need to show a significant difference above the respective background levels which allows them to be distinguished from noise. So, the differences of the sensitivies of the gas analysers but also the differences in the integration times have an impact on the detection of plumes. Further, the different groups used individual algorithms to detect plumes.

Line 258: Is the deviation a known issue for $SO_2$ calibration gas and is this regularly tested for?

AC: The calibration gases were tested in different ways by the groups to ensure their quality. And ways to ensure the quality of calibration gases and the validation of measurement

instrumentation are presented in this paper. However, the magnitude of deviation has not been experienced before.

Line 260: What kind of correction was applied and how were possible affected data identified?

AC: The calibration history was fully traceable from the calibration log which allowed to identify the affected data. A cross-comparison delivered the data needed for a mathematical correction. The sentence is now updated to: "The affected data were corrected afterwards through recalculation using an updated calibration curve. " (l. 280f)

Line 320: Was there a specific reason the UAVs measured in different distances?

AC: The two drones were operated with a safe operating distance from each other for deconfliction, please see explanation in line 230.

**Technical suggestions**

Add NOx = NO + NO2 where it first appeared in the introduction.

AC: This is adapted according to suggestion. (l. 35)

**References**

Beecken, J., Irjala, M., Weigelt, A., Conde, V., Mellqvist, J., Proud, R., … Duyzer, J. (2019). *Review of available remote systems for ship emission measurements* (No. D2.1). The SCIPPER Project (European Commission - Horizon 2020 No. 814893).
Explicit. (2018). *Airborne Monitoring of Sulphur Emissions from Ships in Danish Waters—2017 Campaign Results*. Ministry of Environment and Food of Denmark. Retrieved from https://www2.mst.dk/Udgiv/publications/2018/04/978-87-93710-00-9.pdf

---

## Author Comment (AC2)

We thank Ward van Roy for the highly valuable and thoughtful review. We think, his comments do improve the paper noticeably.

We refer to the individual comments as follows:

**General comments:**

The manuscript is well written and includes valuable information. The research questions are well defined and the methods are well worked and properly referenced. The results are clearly presented and will enhance the scientific knowledge in the field. Some references to the international regulations (IMO, MEPC) could be added and legal wording can be improved. Make sure to use always the latest regulations or consolidated versions in the references to international regulations and conventions. When using legal references avoid mentioning the pages and avoid duplicating the year (at the end) as this has no added value (e.g. 11.5.1999, p. 13–18 (EN), 1999. this can better be mentioned as: adopted on 11 May 1999.).

Reference and comparison to previous studies and literature is somehow lacking. The number of references is relatively short. In the discussion some considerations could have been made on NOx regulations. As all remote monitoring groups measured NOx as well, it would be interesting to see the intercomparison of the NOx measurement results.

The importance of the results for operational compliance monitoring/enforcement could be more elaborated. How did the research contribute to the enforcement practices in the field? What are the lessons learned for the compliance monitoring organizations. Will reporting thresholds be lowered … ?

The evaluation is now limited to the analysis of the absolute differences between the measurements and the BDN/Fuel samples. As this will work fine for the range 0-0.1% FSC. This is less useful for high measurements e.g. non-compliant vessels or measurements outside the ECA. An elaboration of the proportional bias beside the absolute bias would therefore be interesting. If the authors consider that this is outside the scope of this research, the authors should mention this.

AC: The suggested references are now included and suggestions regarding legal referencing are now adapted. There is a practical limitation regarding writing exact dates without doubling the year due to the required reference style by the publisher.

Indeed, emissions of NOx and particulate matter were also measured during the same measurement campaign. The results are foreseen to be published in separate papers but cannot yet be referenced.

The focus of this paper is on state-of-the-art monitoring. The focus of the results is on the intercomparison performance to accurately determine the FSC from plume emission measurements between the currently used instruments within the SECA region. The aim is to assess the capability of the current instrumentation to differ between compliant and non-compliant measurements in this region.

The results of this study were taken at the currently applied sulphur limit for seagoing vessels. For low FSCs, the capability of state-of-the-art instruments that measure VMRs at trace level to quantify the correspondingly low VMRs is typically at its limits. In this case, the relative uncertainties are high

compared to cases that can be expected when ships that go with fuel with sulphur contents closer to 0.5 %Sm/m are measured - assuming comparable measurement conditions.

The limitation has now clearly been mentioned in the introduction (l. 77f).

**Detailed comments:**

15. I Would suggest to use "seagoing" (withthout "-") vessels or use "ocean going vessels"

    AC: This is adapted according to suggestion.

18. suggest rephrasing "observe the same emission sources under similar conditions" as measurements were conducted to measure the emission factors, so I would suggest to mention that.

    AC: This is adapted according to suggestion.

21. Would suggest to stick to the official term "North Sea Sulphur Emission Control Area"

    AC: This is adapted according to suggestion.

24 "reference" without "s"

    AC: This is adapted according to suggestion.

24. Specify "most"

    AC: This is adapted to: "Seven out of the eight".

26. would suggest to split up "the lowest systematic deviation was observed for the airborne system.. having a deviation of … %". "..The lowest the total uncertainty was observed for the laser …"

    AC: This text passage is now restructured.

30. Maybe make a reference to the formerly established thresholds at 95% CI (e.g. 0.2%FSC with 95% CI) to demonstrate the advancement.

    AC: The aim of this study was not to improve the individual uncertainty estimations but to assess the reliability of the individually reported FSC estimations including their uncertainties to make them comparable between the groups.

32. The correct reference to MARPOL is: "International Convention for the Prevention of Pollution from Ships"

Add reference:

International Convention for the Prevention of Pollution from Ships 1973, as modified by the

Protocol of 1978 relating thereto (adopted 17 February 1978 (MARPOL), in force 2 October 1983) 1340 UNTS 61, as amended

AC: This is adapted according to suggestion.

35. Suggestion to add Regulation nrs (Reg. 13 and Reg. 14) here.

AC: This is adapted according to suggestion.

39. Suggestion to add guidelines on scrubbers:

2021 Guidelines for Exhaust Gas Cleaning Systems, Resolution MEPC.340(77), Adopted on 26 November 2021

AC: This is adapted according to suggestion.

41. replace "implemented" by "entered into force" as this are 2 different things. Add a reference:

Protocol of 1997 to amend the International Convention for the Prevention of Pollution from Ships of 2 November 1973, as modified by the Protocol of 17 February 1978,  London, 1997, into force on 19 May 2005 and amended

AC: This is adapted according to suggestion.

43. Make reference to the revised Sulphur Directive of 2016:

Directive (EU) 2016/802 of the European Parliament and of the Council of 11 May 2016 relating to a reduction in the sulphur content of certain liquid fuels, (codification), OJ L 132/58, 21.05.2016

AC: This is adapted according to suggestion.

45. Use the correct legal wording "..North Sea SECA came in effect on 22 November 2007 …" "..Baltic Sea SECA came in effect on 19 May 2006 .." + add reference

List of Special Areas, Emission Control Areas and Particularly Sensitive Sea Areas, Circular MEPC.1/Circ.778/Rev.3 of 2 July 2008

AC: This is adapted according to suggestion.

52. Or quick sampling methods (e.g. XRF)

AC: This is adapted according to suggestion.

70. Specify "near shore" as airplanes can also operate near shore as proven by the BE measurements inside the Westerschelde and as close to the ports as 200m. Suggestion to skip this sentence as it implies that airplanes cannot operate near shore.

AC: This is rephrased now: "Measurements by aircraft and UAV systems can be conducted for monitoring of vessels at any reachable location."

73. Use SCIPPER abbreviation after the written out acronym instead of before.

AC: This is adapted according to suggestion.

148. Provide the manufacturer and model of the EC sensors of the UAS systems like was done for the stationary systems.

AC: Both drone systems are integrated systems. The Explicit system is referred to as the Explicit Mini-Sniffer System. Chalmer's system is experimental and does not have a particular denotation.

149. Ambient air quality laboratory use more stringent calibration methods, why are these not applied here? Not only the sensors need to be calibrated but the VMRs inside the cylinders need to be checked too. Without this step the calibrations risk to be erroneous.

AC: We agree, and this study highlights the importance of the accuracy of the calibration gases and recommends validations of delivered cylinders in reference laboratories amongst other measures to assure quality, e.g round-robin tests. The text is now updated accordingly (l. 279ff).

224. However kn is also correct the use of kts is more common in the maritime sector

AC: This is adapted according to suggestion.

257. Cfr comment 149, issue with unreliable VMRs of ordered gasses is a bit lost here, suggestion to move this to 149. However the data correction for the measurements that were conducted using the sensors calibrated using the wrong VMR reference ratios should be elaborated here.

AC: This section is based on an observation that we made in connection with the calibrations. Therefore, we think, that it should be associated with the discussion section.

264. Please elaborate on possibility that FSC could potentially change between measurement and sampling. My opinion is that for most cases no substantial change is expected, but it can not ruled out, certainly when vessels have been doing fuel change over procedures. A potential may to exclude this potential error, was to limit the analyses to vessels with only 1 fuel type on board.

AC: We agree that theoretically it could be possible that the fuel was switched shortly before or between remote measurement and onboard inspection including fuel sample/collection of bunker delivery note. However, as this is an unlikely case this close to the harbour, we decided to keep all cases in the study. Nevertheless, we clarified this in the text (l. 289ff).

290. The graph is interesting, however, not that relevant for operational use. What is important is not the reported uncertainty, but the maximum allowable difference. For measurements under the non-compliance threshold the reported uncertainty and maximum allowable difference can be the same, however as soon as the non-compliance reporting threshold is reached, the reported uncertainty is irrelevant for operational reasons, but its rather the difference between the FSC measurement and the FSC limit which must be examined, to avoid type II errors where compliant vessels are assigned as non-compliant.

Please also refer to the comparison analysis made by airborne measurement and EGCS/fuel samples

https://www.mdpi.com/2073-4433/14/4/623

AC: Figure 2 shows that deviations between the estimated FSC and the expected FSCs from fuel sampling and bunker delivery notes are off up to a multiple of the currently reported uncertainties.

This is due to the hitherto used error budgeting by the individual groups and proves the value of comparisons in the field in a true-world environment. This is a relevant outcome of this campaign.

The authors intended to elaborate more on the distribution of the absolute deviations over a span of different expected FSCs. However, the values of the expected FSC was heavily centred on 0.082%Sm/m with nearly 70% of all values being in the tight range between 0.075 and 0.010%Sm/m. We added this information to section 2.6 *Fuel samples and bunker delivery notes*.

The suggested reference found surprisingly low uncertainties for a similar system especially for FSCs around 0.1%$S_{m/m}$. We are uncertain if the findings are comparable.

300. Reference could be made to the negative bias correction by Van Roy et al 2022 b which observed similar negative biases.

    AC: The reference is now included in section 3.2 where the negative bias and suggestions for corrective measures are discussed.

357. The over NO correction could also be a potential cause for the negative bias. Not that Van Roy et al. b use lower NO correction factors compared to Chalmers.

    AC: This is discussed in more detail now (l. 381ff).

330. Note that Van Roy et al. evaluated RH and T impact (https://doi.org/10.3390/atmos14040623), see supplementary material.

    AC: In their study, van Roy et al. saw only little dependence on RH and T. Also, between the systems in this study were systems, e.g. exp.uas, that did not indicate such a correlation while it was more apparent for others, e.g. cha.las. As discussed in section 3.2, we suggest that the cause for the observed differences, also with respect to the observation by van Roy, might be in the individual instrumental setups instead of the underlying methodologies. We see a need for further testing on instrumental basis to understand this effect better with a particular focus on areas, on which condensation can occur, like filters. This is already emphasized in the paper (l 369ff).

420. Round robin test are first mentioned here, they should have been mentioned before in the discussion on how to correct negative biases. Suggestion to add reference: https://www.mdpi.com/2073-4433/14/6/969

    AC: This is now explicitly included (also the suggested reference) in the discussion section in connection with the validation of calibration: "This highlights the need to validate VMRs of the calibration gases. Possible ways to conduct such a validation are by tests against the preceding calibration gases or, with higher accuracy, by accredited reference laboratories. Alternatively, round-robin tests can be used to validate the instrument calibration using reference gases or gas blends simulating different FSCs (Van Roy et al., 2023)." (l. 277ff).